

**Artificial Radionuclides in Squid from northwestern Pacific in 2011 following the**

**Fukushima accident**

Wen Yu[1], Mathew P. Johansen[2], Jianhua He[1], Wu Men[1], Longshan Lin[1]

*1 Third Institute of Oceanography, State Oceanic Administration of China, Xiamen, China*

*2 Australian Nuclear Science and Technology Organization, Sydney, Australia*

**Abstract**:

In order to better understand the impact of Fukushima Nuclear Power Plant (NPP) Accident on

commercial marine species, squid (*Ommastrephe bartrami*) samples, obtained from the

northwestern Pacific in November 2011, were analyzed for a range of artificial and natural

radionuclides (Cs-134, Cs-137, Ag-110m, U-238, Ra-226 and K-40). Short-lived radionuclides Cs-

134 and Ag-110m released from Fukushima NPP Accident were found in the samples, with an

extremely high water-to-organism concentration ratio for Ag-110m (> 2.9E+04). The radiological

dose rates for the squid from the radionuclides measured were far lower than the relevant benchmark

of 10 $\mu$Gy h$^{-1}$. For human consumers ingesting these squid, the dose contribution from natural

radionuclides (>99.9%) including Po-210, was far greater than that of Fukushima-accident

radionuclides (<0.1%). The whole-body to tissue and whole-body to gut concentration ratios were

calculated and reported, providing a simple method to estimate the whole-body concentration in the

environmental monitoring programs, and filling the data gap of concentration ratios in cephalopods.

Our results help fill data gaps on uptake of NPP radionuclides in the commercially important

Cephalopoda class and add to scarce data on open-ocean nekton in the northwestern Pacific soon

after the Fukushima accident.

**Key words**:

Fukushima NPP Accident; squid; concentration ratios; radiological dose; Silver-110m.

**1.    Introduction**

The Fukushima Daiichi Nuclear Power Plant (NPP) Accident, which was caused by the combined

effect of the great earthquake and subsequent tsunami in March 2011, resulted in increased levels





of artificial radioactivity in the marine environment to the east of Japan (IAEA 2015). The

radioactive releases, dominated by radiocesium, were transported and dispersed widely in the North

Pacific within a few years (Aoyama et al., ;Smith et al.), raising concerns about the potential impact

on the marine biota and human consumers of seafood products.

A large amount of research has been conducted to determine the level of artificial radionuclides in

biota samples and to assess the relevant radiological impact to both human and marine species.

However, most studies have focused on the concentration of radiocesium in fish (Johansen et al.,

2015), and only a few publications have reported on radionuclides in other marine species

(Buesseler et al., ;Yu et al.). Few data are available for open-ocean locations as compared with

coastal areas, especially from 2011. Filling these data gaps will improve and expand understanding

of the dynamics of cesium in the early months following the accident.

*Ommastrephe bartrami* (neon flying squid) is a migratory squid species that is commercially

important, consumed by humans, and common in the Pacific Ocean and circumglobally in temperate

and tropical waters. It feeds near the surface on small fish and is thus a potential accumulator of

radiocesium via diet and water pathways. Moreover, cephalopods have a strong capability to

accumulate silver in their bodies (Miramand and Bentley, 1992;Bustamante et al., 2004) and would

potentially indicate uptake of the short-lived (0.70 year half-life) Ag-110m released from Fukushima

Daiichi NPP Accident. Similarly, the presence of Cs-134 (2.1 year half-life) in samples would also

indicate a pathway from Fukushima Daiichi NPP releases. Therefore, specimens captured at

locations in the North Pacific may serve as bio-indicators of the presence, strength, and movement

of the radioactive signal from Fukushima Daiichi Accident.

This study assesses samples of *O. bartrami* obtained from the northwestern Pacific in November

2011 for a range of artificial and natural radionuclides (Cs-134, Cs-137, Ag-110m, U-238, Ra-226

and K-40). The radiological dose rates and relevant risk levels were determined for the squid, as

well as potential dose rates to human consumers of squid seafood. Consistent with international

efforts to compile transfer data, Concentration Ratios (wholebody-to-water and wholebody-to-tissue)

are calculated and reported, including those for different age classes of squid.



## 2.  Materials and methods

### 2.1.  Sample collection and analytical procedure

Thirteen composite samples of *O. bartrami* samples with a total weight of 126.2 kg were obtained by bait fishing in open water in northwestern Pacific. Six sampling locations were selected in the area of 34°-39°N to 145°-149°E to investigate eastward deposition and oceanic migration pathways of radionuclide releases from the Fukushima Daiichi NPP (Fig 1). To ensure sample mass was sufficient to reach minimum detectable activity (MDA) levels for key radionuclides, composite samples were made with multiple specimens from the same sampling site. For those sites with enough sample mass, the specimens were divided into different composite categories according to their body weight. Specimens with body mass less than 1 kg were categorized as "small", those between 1 kg and 2 kg were categorized as "medium" and those more than 2 kg were categorized as "large". The samples were -18 °C frozen on board for transport to the laboratory for the subsequent analysis.

Squid samples were dissected into muscle and gut tissues after thawing, dried at 50 °C, and ashed at 450 °C. The fresh weight and ash weight of the composite samples were recorded. The ash was sealed in cylindrical 75 mm diameter containers, and then subjected to HPGe spectrometry for detection of gamma-emitting radionuclides.

Gamma rays from artificial radionuclides (Cs-134, Cs-137, Ag-110m, Co-58, Co-60, Mn-54 and Zn-65) and natural radionuclides (K-40, Ra-226 and U-238) were analyzed using a planar high-purity germanium (HPGe) detector (Model BE6530 with Multi Channel Analyzer Lynx system; Canberra, U.S.A.). Detection efficiencies for the geometry used were 2.7885 % and 2.4476 %, for Cs-137 and Ag-110m respectively. The counting time for each sample was 24 hr. Genie 2000 software was used to analyze the respective peaks in the energy spectrum. The concentrations were corrected for decay to the initial date of the nuclear accident on 12 March 2011, when the first hydrogen explosion occurred in Unit 1 of the FDNPP (Wakeford).

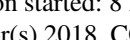


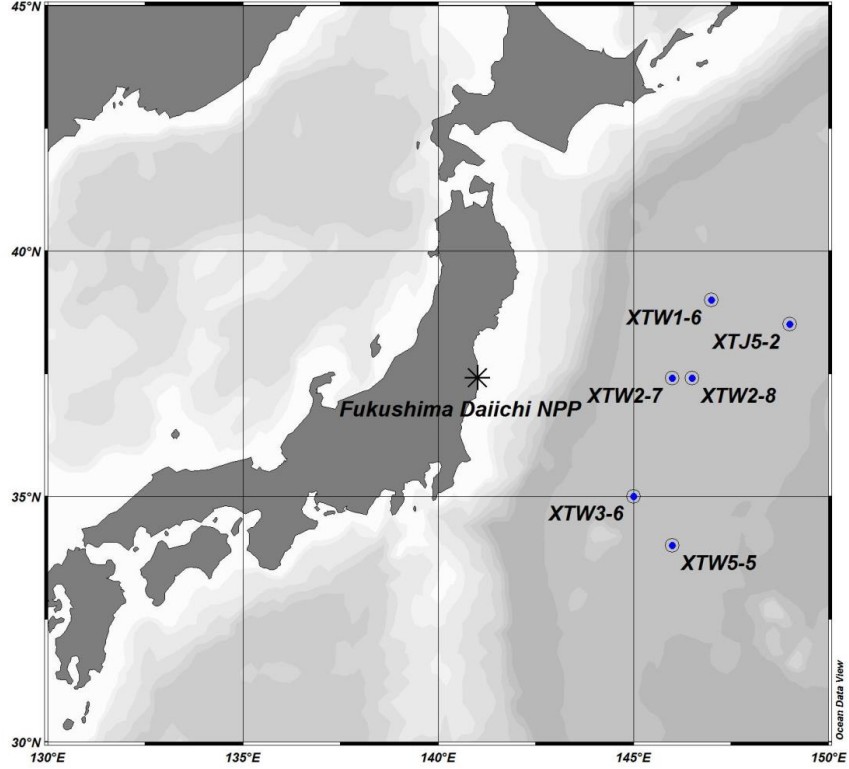

Fig 1 Map of sampling sites

## 2.2. Dose assessment for squid

The ERICA Assessment Tool (version 1.2) (Brown et al., 2008) was used with Tier 2 assessment
to evaluate the radiological risk to squid from the study areas in 2011.The ERICA Tool includes
the capability to specify organism sizes, and in this study, average mass (1.3 kg) and dimensions
(ellipsoid equivalent of 0.3m, 0.1m, 0.085m length, width, height respectively) from the
specimens were used to calculate dose rates. The dimensions of the average *O. bartrami* happen
to be very similar to the standard ERICA "pelagic fish" and therefore the dose rates are very similar
as calculated by ERICA. The measured activity concentrations in the whole-body of $^{137}$Cs, $^{134}$Cs,
$^{110m}$Ag, $^{226}$Ra and $^{238}$U in the samples were used as dose calculation input. The maximum tissue
activity concentrations were used for a more conservative result. As *O. bartrami* are migratory,
their radionuclide tissue levels represent an integrated accumulation from recently traversed
areas in the open ocean area. The exact migratory routes are not known. Therefore, the external
dose rates to the squid were calculated using the average of water radioactivity levels in the



study capture region (average of samples across all sampling locations). Use of the average is

reasonable in this instance as the external dose rates for artificial radionuclides were much

smaller than internal dose rates and therefore variable water activity concentrations had little

influence on overall dose results. For internal dose rates to squid, the dose conversion

coefficients (DCCs) were calculated within the ERICA tool (supplemental). The occupancy

factors were 100% in water, and weighting factors of internal low beta, internal beta/gamma

and internal alpha were set as 3, 1 and 10 respectively.

**2.3.    Dose from ingesting squid by human consumers**

Committed effective doses (Sv) to human consumers of squid were estimated using standard

exposure-to-dose conversion factors for ingestion from ICRP Compendium of dose coefficients

based on ICRP Publication 60 (ICRP, 1999). Key DCFs are 1.30E-08 and 1.90E-08 Sv Bq$^{-1}$ for Cs-

137 and Cs-134 (DCFs provided in the supplemental). The factors are multiplied by intake (e.g. kg

110   yr-1) to obtain committed effective doses to the consumer. In this study, the annual intake rate of

seafood by an adult consumer is assumed to be 20 kg yr$^{-1}$ (consistent with world per capita fish and

related seafood consumption (FAO, 2016). As a conservative assumption, the entire 20 kg yr$^{-1}$ for a

hypothetical consumer is assumed to be sourced from the squid of the study area east of the

Fukushima Daiichi NPP (in practice, only a small percentage of seafood diet would be sourced from

this region). As most dose to human consumers of seafood typically comes from the natural

radionuclide Po-210 (~89%;(Johansen et al., 2015)), the seafood ingestion dose rates here included

Po-210 using conservative generic data for marine seafood(Carvalho, 2011;Hosseini et al., 2010).

**2.4.    Whole-body concentration ratios**

The water-to-organism whole-body Concentration Ratio $CR_{WB}$ (in L/kg) used here is defined as:

$$CR_{WB:Water} = \frac{Whole-Body\ Activity\ Concentration\ (fresh\ mass)\ (Bq/kg-wet)}{Water\ Activity\ Concentration(Bq/L)}$$    ( 1 )

The whole-body activity of an radionuclide was estimated using a mass balance approach

(Yankovich et al., 2010) to reconstruct the amount of radionuclide in the whole-body of the squid.

The whole-body to tissue concentration ratio (dimensionless) was estimated as:

$$CR_{WB:Tissue} = \frac{\sum[Tissue_t\ Activity\ Concentration\ (fresh\ mass) \cdot Tissue_t\ fresh\ mass\ fraction]}{Tissue_t\ Activity\ Concentration\ (fresh\ mass)}$$    ( 2 )





## 3. Results and discussions

### 3.1. Description of *O. bartrami* specimens

In total, 98 specimens were obtained from 6 stations. The mass of the specimens ranged from 118 g

to 2551 g, on average 1347 g. Sixty percent of the specimens weighed 701 g to 1700 g. The trunk

length of the specimens ranged from 115 mm to 440 mm, on average 333 mm. Seventy-five percent

of the specimens had a length greater than 290 mm (adult size) suggesting that the majority of the

specimens were hatched in winter of 2010 or spring in 2011 and had been living for 8 to 11 months

(Wang and Chen, 2005). Combining the estimated age of the squid, and assuming residence in the

general area east of Fukushima Prefecture, it can be inferred that most specimens had been

accumulating radionuclides since Fukushima Daiichi NPP accident, while a minor proportion (the

small size category) were likely to have been hatched after the accident and had shorter exposure

times.

### 3.2. Activity concentrations and CRs in squid

The activity levels of radionuclides in Table 1 indicate that all *O. bartrami* size classes had

accumulated radionuclides from Fukushima Daiichi NPP releases as indicated by Cs-134 and Ag-

110m. The squid specimens had a strong capability to concentrate Ag in their bodies. The maximum

activity of Ag-110m in the whole body of *O. bartrami* was up to 9 Bq/kg, as compared to that in

water which was below the MDA of 0.22 Bq/m$^3$, indicating a maximum concentration factor that is

higher than $4\times10^4$. The mean CRs for Ag-110m were calculated as $>2.95\times10^4 \pm 9.84\times10^3$ (Table 2),

using the MDA as the activity of seawater in Equation (1).

Although this estimate is with large uncertainties because of using MDA of Ag-110m as the water

concentration, these Ag data provide new insights for international researchers and fill a gap as the

relevant international database (Wildlife Transfer Parameter Database;

www.wildliftransferdatabase.org) which has entrees for Ag uptake in the mollusk category, but none

specifically for squid/cephalopods.

The mean $CR_{WB}$ values for Cs-134 and Cs-137 in *O. bartrami* were 6.33 (± 2.80 S.D.) and 5.57

(±2.59 S.D.) respectively. These values are similar to previously published mean concentration




factors for Cs in cephalopods ranging from 9 to 14 (IAEA, 1978;Ishii et al., 1978;Suzuki et al., 1978;IAEA, 2004). The slightly lower $CR_{WB}$ in this study is well within the range of expected variation, which can be very high for water-to-organism CR values (e.g. reported CRs for Cs-137 in marine fish range over nearly an order of magnitude) (Beresford, 2010). The activity concentration of $^{137}$Cs/$^{134}$Cs in the research area reached a maximum of ~600 Bq m$^{-3}$ in June 2011 and soon decreased to below 100 Bq m$^{-3}$ (Aoyama et al., 2016). Considering the temporal change of radiocesium in seawater and its relatively short biological half-life (~70 days) in marine organisms, in this study, the CR calculation used mean Cs-134 and Cs-137 seawater activity concentrations (35.1 and 36.2 Bq m$^{-3}$ respectively) from this study's November 2011 sampling, which were similar to the ~50 Bq m$^{-3}$ reported for July-December timeframe from the same open ocean area (Kaeriyama, 2017).

The results also showed that both Cs-134 and Cs-137 were concentrated mainly in the muscle of the squid. Cesium behaves similarly to potassium in biota and it tends to be distributed to the muscle tissue. These results for the open ocean, real-world conditions are consistent with previous laboratory results of 80+ % accumulation in the muscle and head of cuttlefish after only 8 hours of exposure to water (Bustamante et al., 2004). In contrast, for Ag, the open ocean squid had 95 % Ag in the gut vs muscle.   This result was also consistent with the laboratory cuttlefish which had 98% Ag in the gut following a single spiked feeding and 29 d depuration (Bustamante et al., 2004). From the same study, within the gut, accumulation of Ag is dominantly in the digestive gland.

The smallest squid samples had the highest concentration factors for Cs-134, Cs-137, Ag-110m and U-238 (Fig 3). The higher accumulation occurred in the smaller size class despite their inferred shorter exposure times (shorter lifespan) as compared with the larger size class. These results are consistent with observed Cs depuration rates in juveniles cephalopods (*Sepia officinalis)* being ~four times slower than that of adults, with however, both being relatively fast (adult cuttlefish biological half-life of 16 days for Cs and 9 days for Ag (Bustamante et al., 2004)). This previous study, suggests the radiocesium accumulation and depuration in *O. bartrami* is relatively rapid and that our results therefore primarily reflect recent (~ several months) exposure rather than longer-term accumulation.



The levels of activity for $^{58}$Co, $^{60}$Co, $^{54}$Mn and $^{65}$Zn in the samples were all below the MDA (0.22

182     mBq/g-ash).

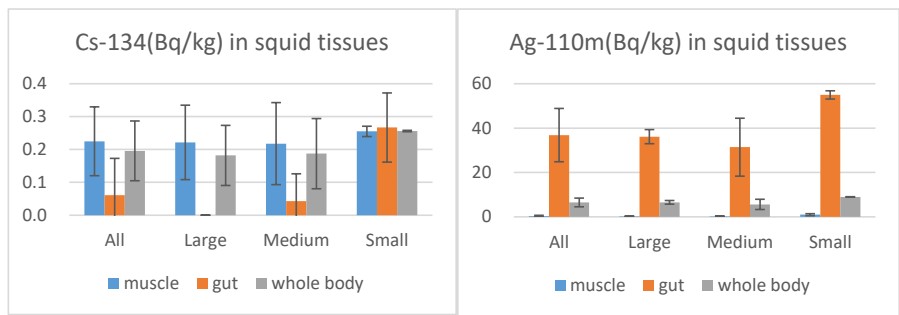

Fig 2 Activity concentrations of Cs-134 (left) and Ag-110m (right) in squid tissues.



Table 1 Statistics of radionuclides' levels in composite samples (Bq/kg-fresh mass)

| Size | Tissues | Cs-137 | | Cs-134 | | Ag-110m | | K-40 | | Ra-226 | | U-238 | |
|---|---|---|---|---|---|---|---|---|---|---|---|---|---|
| | | Range | Average | Range | Average | Range | Average | Range | Average | Range | Average | Range | Average |
| All (n=13) | M | 0.10-0.46 | 0.27±0.12 | 0.06-0.39 | 0.22±0.10 | 0.06-1.29 | 0.36±0.33 | 56.29-94.80 | 76.05±9.40 | nd-0.07 | 0.03±0.03 | 0.16-1.77 | 0.59±0.44 |
| | G | nd-0.33 | 0.05±0.10 | nd-0.34 | 0.06±0.11 | 8.10-56.27 | 36.85±12.02 | 9.72-72.37 | 53.03±15.77 | nd-0.89 | 0.28±0.28 | nd-26.89 | 5.40±7.60 |
| | WB | 0.08-0.38 | 0.23±0.10 | 0.05-0.31 | 0.20±0.09 | 1.70-9.04 | 6.49±1.97 | 53.67-88.09 | 72.13±8.45 | nd-0.17 | 0.07±0.05 | 0.27-5.32 | 1.35±1.40 |
| Large (n=5) | M | 0.13-0.46 | 0.26±0.13 | 0.09-0.39 | 0.22±0.11 | 0.06-0.36 | 0.24±0.13 | 67.62-94.80 | 80.55±11.49 | nd-0.07 | 0.04±0.03 | 0.33-0.94 | 0.63±0.26 |
| | G | ND | ND | ND | ND | 32.25-40.50 | 36.14±3.19 | 49.57-58.88 | 53.00±3.66 | nd-0.68 | 0.24±0.27 | nd-7.89 | 2.61±3.22 |
| | WB | 0.11-0.38 | 0.21±0.11 | 0.08-0.31 | 0.18±0.09 | 5.53-7.78 | 6.54±0.83 | 64.55-88.09 | 75.76±9.86 | nd-0.15 | 0.07±0.06 | 0.54-2.09 | 0.97±0.63 |
| Medium (n=6) | M | 0.10-0.41 | 0.27±0.14 | 0.06-0.34 | 0.22±0.12 | 0.06-0.46 | 0.25±0.19 | 56.29-78.78 | 72.30±8.12 | nd-0.05 | 0.02±0.02 | 0.16-0.86 | 0.40±0.28 |
| | G | nd-0.13 | 0.03±0.05 | nd-0.21 | 0.04±0.08 | 8.10-45.85 | 31.40±13.05 | 9.72-67.73 | 47.80±21.18 | nd-0.53 | 0.18±0.19 | nd-10.44 | 3.05±3.84 |
| | WB | 0.08-0.35 | 0.23±0.12 | 0.05-0.29 | 0.19±0.11 | 1.70-8.09 | 5.61±2.31 | 53.67-73.94 | 68.16±7.56 | nd-0.11 | 0.04±0.04 | 0.27-2.45 | 0.86±0.80 |
| Small (n=2) | M | 0.21-0.34 | 0.27±0.09 | 0.24-0.27 | 0.25±0.02 | 0.65-1.29 | 0.97±0.45 | 73.26-78.88 | 76.07±3.97 | nd-0.05 | 0.02±0.03 | 0.41-1.77 | 1.09±0.97 |
| | G | 0.20-0.33 | 0.27±0.09 | 0.19-0.34 | 0.27±0.11 | 53.64-56.27 | 54.95±1.86 | 65.30-72.37 | 68.83±4.99 | 0.44-0.89 | 0.67±0.32 | 12.00-26.89 | 19.45±10.53 |
| | WB | 0.21-0.34 | 0.27±0.09 | 0.25-0.26 | 0.26±0.00 | 8.90-9.04 | 8.97±0.10 | 73.13-76.76 | 74.95±2.57 | 0.07-0.17 | 0.12±0.07 | 2.21-5.32 | 3.76±2.19 |

* Tissues: M – muscle, G – gut, WB – whole body. **ND: level was below the minimum detectable activity.



### 3.3. Whole-body:muscle and whole-body:gut concentration ratios

Most of non-human biota radiation dose assessing models focus on estimation of dose rates using the *whole-body* activity concentrations of radionuclides (Brown et al., 2008;DOE, 2004). However, muscle tissue (vs. whole-body) is measured in most monitoring programs which typically focus on seafood tissues consumed by humans. Therefore, there exists a need for whole-body:tissue concentration ratios that allow for estimation of whole-body concentrations from commonly measured tissue data (Yankovich et al., 2010).

The whole-body:muscle and whole-body:gut concentration ratios for radionuclides in squid samples are listed in Table 2. For many radionuclides, the tissue-specific concentration values for the small squids tend to be higher than those for large squids. The uncertainty of the whole-body:gut CRs for Cs-137 and Cs-134 are relatively high because of the relatively low level and large activity range of radiocesium in the gut samples. These CRs presented here are calculated for the non-equilibrium conditions following the accident. This issue is somewhat compensated for by focusing on radionuclides that are taken up relatively quickly, and by using the average activity concentrations over their relatively short lifespan of the squid. Equilibrium conditions are generally not achieved in natural systems, and our results, like all CRs should be considered in context. Further research is necessary to obtain a better estimation the biokinetics of uptake in squid and of the whole-body:gut CRs for Cs-137 and Cs-134.





Table 2 Concentration ratios for radionuclides in 2011 conditions following the accident (see text).

| CR* | Size | Cs-137 | Cs-134 | Ag-110m | K-40 | Ra-226 | U-238 |
|---|---|---|---|---|---|---|---|
| WB-M | All | 0.93±0.28 | 0.94±0.30 | 41.87±39.49 | 1.04±0.28 | 2.75±1.60 | 2.36±1.36 |
| | Large | 0.82±0.01 | 0.82±0.01 | 38.89±30.21 | 0.94±0.01 | 2.42±1.69 | 1.64±0.79 |
| | Medium | 0.85±0.03 | 0.86±0.06 | 47.90±50.29 | 0.96±0.01 | 2.00±1.53 | 2.35±1.20 |
| | Small | 1.00±0.00 | 1.01±0.07 | 10.30±4.67 | 0.99±0.02 | 3.58 | 4.22±1.73 |
| WB-G | All | 2.59±2.50 | 2.29±2.44 | 0.18±0.02 | 1.30±0.18 | 0.33±0.29 | 0.24±0.03 |
| | Large | NA** | NA** | 0.18±0.01 | 1.43±0.18 | 0.50±0.43 | 0.28±0.01 |
| | Medium | 4.15±4.15 | 3.54±3.54 | 0.18±0.02 | 1.25±0.12 | 0.24±0.09 | 0.24±0.01 |
| | Small | 1.03±0.01 | 1.04±0.40 | 0.16±0.00 | 1.09±0.12 | 0.17±0.02 | 0.19±0.01 |
| WB-W*** | All | 6.33±2.80 | 5.57±2.59 | >2.95E+4 ± 8.94E+3 | 6.17±0.71 | 14.66±11.92 | 37.97±39.39 |
| | Large | 5.90±2.91 | 5.18±2.60 | >2.97E+4 ± 3.76E+3 | 6.42±0.84 | 16.35±12.46 | 27.34±17.89 |
| | Medium | 6.30±3.17 | 5.33±3.04 | >2.55E+4 ± 1.05E+4 | 5.89±0.69 | 9.56±9.40 | 24.11±22.43 |
| | Small | 7.52±2.52 | 7.29±0.06 | >4.08E+4 ± 4.50E+2 | 6.35±0.22 | 25.76±15.07 | 106.09±61.85 |

* CR: WB-M is whole-body to muscle concentration ratios, WB-G is whole-body to gut concentration ratios, and WB-W is whole-body to water concentration ratios.

** NA: Data is not available because radioactivity of specific radionuclides in at least one tissue was below MDA.

*** Values were calculated using mean Cs-134 and Cs-137 seawater activity concentrations of 35.1 and 36.2 Bq m$^{-3}$ , and the MDA of $^{110m}$Ag in seawater (0.22 Bqm$^{-3}$).





### 3.4. Dose assessment results

### 3.4.1. Dose rates for squid

The internal radiological dose rates to squid from artificial radionuclides ($^{110m}$Ag, $^{134}$Cs and $^{137}$Cs) were collectively much higher than the external dose rates (Fig. 4). This is consistent with the observed accumulation of radionuclides inside the squid body as compared with that in the surrounding seawater. The internal dose rates from FDNPP-associated artificial radionuclides were lower, by two orders of magnitude, than those from the natural radionuclides measured in this study. From these radionuclides, only approximately 1.4 % of the total dose rate is estimated to have come from the Fukushima Daiichi NPP releases. The total dose rate for squid is 0.15 µGy·h$^{-1}$ from study radionuclides, and increases to approximately 0.61 µGy·h$^{-1}$ when adding Po-210 a natural radionuclide and significant dose contributor in marine organisms (using a conservative generic marine value of 15 Bq kg$^{-1}$-fresh mass and 0.001 Bq L$^{-1}$ in squid and seawater, respectively based on (Carvalho, 2011) and (Hosseini et al., 2010). These dose rates are much lower than the most conservative screening benchmark dose rate of 10 µGy·h$^{-1}$ (Garnier-Laplace et al., 2008). The dose calculations used the measured activity concentrations in the squid (not CRs) and the calculated dose rates represent a point in time (November 2011) with likely higher doses prior to, and lower doses following the sampling date. However, the relatively low values indicate a more detailed (e.g. pulse-dynamic uptake) dose calculation) is not necessary in this case. Overall, results indicate that the radioactive releases from the Fukushima accident would not have a significant adverse effect on *O.* bartrami individuals or populations living in the study area.




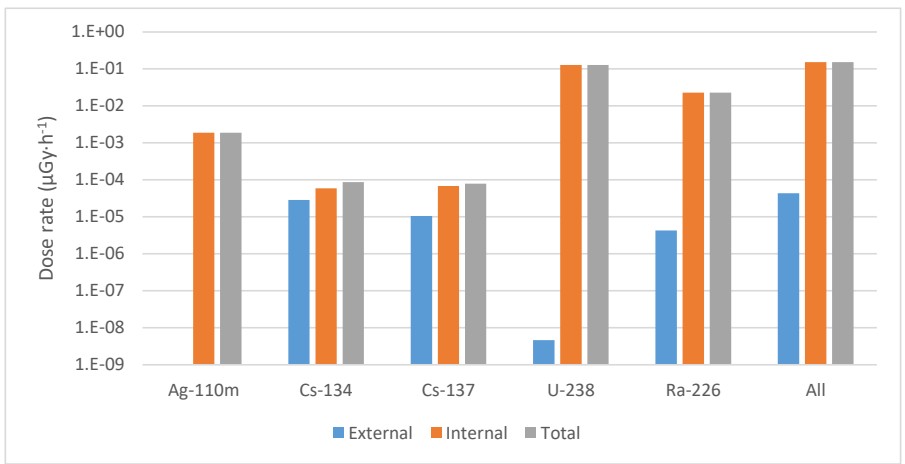

Fig 3 Dose rates (µGy·h⁻¹) from measured radionuclides for squid samples

### 3.4.2. Dose rates for human consumers of seafood

From the radionuclides measured in edible squid tissue (muscle), a committed effective ingestion

dose of 0.01 mSv (median; minimum = 0.007 mSv, maximum = 0.014 mSv) would have occurred

to a hypothetical human consumer of 20 kg yr⁻¹ of squid from the study area (based on squid

captured in November 2011). The doses calculated here are hypothetical and are intended to be

conservative overestimates given the unrealistic assumption that all of the consumer's yearly

seafood came from the study area. If consumption of Po-210 (from natural background) is also

included, the total dose increases to 0.30 mSv, with almost all derived from Po-210 (Table 3). Of

this dose (including Po-210), less than 0.1 % is estimated to have been sourced from the Fukushima

Daiichi NPP. This is consistent with previous findings that natural radionuclides provided far greater

dose rates to potential consumers of Pacific tuna (Fisher et al., 2013), and even for seafood sourced

within a few kilometers of the Fukushima Daiichi NPP in 2013 (Johansen et al., 2015). The dose

contribution from the Fukushima Daiichi NPP releases for squid consumption of this study are far

below the 1 mSv per year recommended constraint for prolonged exposure by the public from

nuclear facility releases (ICRP, 1999).

Table 3. Ingestion dose estimates to human consumers of the squid in this study (Sv y⁻¹ based on

20 kg consumption of study squid).

| | minimum | median | maximum | % this study* |
|---|---|---|---|---|



| | | | | |
|---|---|---|---|---|
| **K-40** | 6.98E-06 | 9.43E-06 | 1.18E-05 | 3.12% |
| **Ag-110m** | 3.36E-09 | 2.02E-08 | 7.22E-08 | 0.01% |
| **Cs-134** | 2.28E-08 | 8.36E-08 | 1.48E-07 | 0.03% |
| **Cs-137** | 2.60E-08 | 7.02E-08 | 1.20E-07 | 0.02% |
| **Ra-226** | | 1.68E-07 | 3.92E-07 | 0.06% |
| **U-238** | 1.44E-07 | 5.31E-07 | 1.59E-06 | 0.18% |
| **Po-210\*\*** | 1.44E-05 | 2.92E-04 | 1.08E-03 | 96.59% |

\* Based on median activity concentration values this study (Table 1 data, average of all sizes).

\*\* Po-210 from generic published data (Carvalho, 2011;Hosseini et al., 2010).

## 4. Conclusions

Elevated levels of Cs-134 and Ag-110m from Fukushima NPP Accident were found in the squid (*O.*

*bartrami*) samples collected at NW Pacific in November 2011. This study filled a gap in

international transfer data by providing concentration ratios for several key NPP-associated

radionuclides in the whole-body and tissues of cephalopods. The Concentration Ratio for Ag-110m

in squid was found as high as $4 \times 10^4$ L/kg in the smallest samples, with a mean value of $2.95 \times 10^4$

L/kg in all the samples, indicating that squid was a good bioindicator for Ag-110m from Fukushima

NPP Accident. The radiological dose contribution from the Fukushima Daiichi NPP releases for

squid living in the study area, and for human consumers of these squid, were both far below the

recommended dose limits. By comparison, natural radionuclides, particularly Po-210, provide

orders of magnitude greater dose rates.

### Acknowledgement

This study was partially supported by the Scientific Research Foundation of Third Institute of

Oceanography, SOA (2015010), the Northwestern Pacific Marine Environmental Monitoring

Project, the International Organizations and Conferences Project of the State Oceanic

Administration of China, the Coordinated Research Project (CRP K41017) and Regional

Cooperative Agreement Project (IAEA/RCA RAS7028) of the International Atomic Energy Agency

(IAEA), and the Public Science and Technology Research Funds Projects of Ocean (201505005-1).

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
