# Peer review of "Artificial radionuclides in neon flying squid from the northwestern Pacific in 2011 following the Fukushima accident"

_Biogeosciences, 2018_

## Referee Comment (RC1) · Anonymous Referee #1 · 29 May 2018

General comments; Temporal change of radionuclides by the Fukushima NPP accident is larger than the ones of released events, therefore it is difficult to adapt the Concentration Ratio (CR). On the other hands, the CR is useful to compare the Fukushima Dai-ichi NPP accident with other previous release events at the specific period. Even if the public did not eat seafood affected by the Fukushima Dai-ichi NPP accident, the authors quantitatively showed that dose rate by the Fukushima Dai-ichi NPP accident, included the Ag-110m data, should be quite smaller than the one by natural radionuclides, Po-210. Therefore, this paper is suitable for the publication of the Biogeosciences with following minor revisions.

Specific comments; Line 30 ; Radionuclides released to the North Pacific due to atmospheric deposition and direct discharge. The authors should describe the transport process in more detail by referring Aoyama et al. (2016).

Line 108 ; Explanation is needed for "DCF".

Line 146 ; I understand that there is no CF data for Ag-110m in squid. On the other hand, IAEA. Technical Reports Series No.422 show the high CF for marine organism. These information is useful for the discussion.

Line 157 ; 137Cs/134Cs -> Cs-137

---

## Author Comment (AC1) · 31 May 2018

We thank the anonymous reviewer for his useful comments. We largely agree with the points raised and considered many of them in the revised version of the manuscript. In the following, our changes are listed next to the points raised.

General comments;

Temporal change of radionuclides by the Fukushima NPP accident is larger than the ones of released events, therefore it is difficult to adapt the Concentration Ratio (CR). On the other hands, the CR is useful to compare the Fukushima Dai-ichi NPP accident with other previous release events at the specific period. Even if the public did not eat seafood affected by the Fukushima Dai-ichi NPP accident, the authors quantitatively showed that dose rate by the Fukushima Dai-ichi NPP accident, included the Ag-110m data, should be quite smaller than the one by natural radionuclides, Po-210. Therefore, this paper is suitable for the publication of the Biogeosciences with following minor revisions.

Specific comments;

Point 1: Line 30 ; Radionuclides released to the North Pacific due to atmospheric deposition and direct discharge. The authors should describe the transport process in more detail by referring Aoyama et al. (2016).

Reply: We agree to this point. The main transport pathway and the estimated transporting speed for surface cesium was added in Line 31-32.

Point 2: Line 108 ; Explanation is needed for "DCF".

Reply: We agreed with this point. The explanation for "DCF", exposure-to-dose conversion factor, was added in Line 108.

Point 3: Line 146 ; I understand that there is no CF data for Ag-110m in squid. On the other hand, IAEA. Technical Reports Series No.422 show the high CF for marine organism. These information is useful for the discussion.

Reply: We agreed with this point. Relevant information of CF for Ag in molluscs was added in Line 151-152.

Point 4: Line 157 ; 137Cs/134Cs -> Cs-137

Reply: We agreed with this point. The text "137Cs/134Cs" was changed into "137Cs" in Line 160.

Please also note the supplement to this comment:

[Figure]

https://www.biogeosciences-discuss.net/bg-2018-133/bg-2018-133-AC1-supplement.pdf

[Figure]

**Supplement:**

[revised manuscript text omitted]

---

## Referee Comment (RC2) · Anonymous Referee #3 · 10 Jul 2018

The study investigates the radionuclides (both natural and artificial ones) in the neon flying squids from east Japan following the Fukushima disaster in 2011. It has merits and deserves publication but some points need to be modified before its acceptance. In particular, the calculations of internal doses and human exposure for polonium should be based on studies dedicated to squid as well to avoid biased estimations (see below). Also, the ms should have been prepared with more care as there are many mistakes all along the text which should have been avoided by a careful reading.

Specific points:

The title suggest a general approach on "squid" but only one single species is used

in the study. I suggest to modify the title by including the name of the squid as follow "Artificial radionuclides in the neon flying suid Ommastrephes bartramii from. . ..."

Line 10. The correct name of the species is Ommastrephes bartramii. This is to be changed consistently throughout the ms. Also specify here "neon flying squid" Line 14. It should be easier for the readers to write 2.9 104. This should be used consistently throughout the ms. Line 22. I agree that cephalopods constitute an important commercially group but here you considered only one single species and it may be somewhat tricky to extrapolate the present result to the whole group, especially to nectobenthic species (cuttlefishes) and to coastal benthic ones (octopuses). Do you believe that similar results are to be found for such Orders? Line 32 and Line 38. The years are missing for the references. Line 38. M&M. Where the sexes considered when grouping the individuals? Sexual dimorphism occurs in this species so it can results in grouping individuals of similar size/weight but with different ages. How did you manage this? Line 72. Gut tissues is very vague and seems to mainly refer to organs and tissues involved in the digestive processes. If this is true, it means that other tissues such as the gills, heart, gonads and associated glands were not considered. Can you please clarify? Line 74. Define HPGe here and remove it at Line 78. Lines 79-80. Detection efficiencies for the other radionuclides should be also provided here. Line 112. "yr-1" Line 118 and Line 119. Spaces are missing before and inside the references. Please prepare you ms with more care. Line 124. "activity of a radionuclide" Line 127. As for CRWB:water, define CRWB:Tissue Lines 130-140. This paragraph should move to the M&M section: it is not "results" but just a description of the sampling which was missing in the M&M section. Page 6. The table is a duplicate of Table 1 page 9. Remove it from page 6. Line 147. Do you mean independently of the size classes? Line 160. CF factor has been determined experimentally for cuttlefish by Bustamante et al. 2006 in JEMBE with lower values than reported here. Line 171. Change Bustamante et al 2004 (dedicated to Ag and Co) by Bustamante et al. 2006 (dedicated to Cs and Am). Line 174. One important aspect is that the digestive gland is the storage tissue independently of the exposure pathway (food or seawater). Line 175. Is this significant?

Line 179. Provide a reference. Line 180. Add Bustamante et al. 2006 as a reference for Cs. Page 9, Table 1. "Statistics" in the title is not appropriate here; there is no statistics in this table but activities of the radionuclides only. For "small individuals", means and standard deviation have been calculated with only 2 individuals, which is not fully correct. Page 11. *** is not applied to Cs, so it should be limited to Ag. Line 216. The value of 15Bq/kg seems a bit high compare to what it is found for muscle in squids. In the cited review (Carvalho 2011) , the value is 1.61 Bq/kg wwt, so I guess you took the wrong value in the table. See also for example Waska et al 2008 in STOTEN who reported 5.7 Bq/kg dry wt (so approx. 5 times less when expressed relatively to the fresh weight) in the squid Todarodes pacificus from the Japan Sea. Also, Heyraud et al. 1994 reported values of 15 to 21 Bq/kg dry wt (so between 3 to 4 in wet wt) in Loligo vulgaris from South Africa. Revise your dose calculation accordingly. Line 231. Do you mean "0.010 mSv" ? Line 234-243. Calculations to be revised according to relevant Po values. References. The bibliographic references should be homogeneous. For example, Line 276, the journal title is not in full as for the other references.

---

## Author Comment (AC2) · 1 Aug 2018

Referee #3 comments and responses:

The study investigates the radionuclides (both natural and artificial ones) in the neon flying squids from east Japan following the Fukushima disaster in 2011. It has merits and deserves publication but some points need to be modified before its acceptance. In particular, the calculations of internal doses and human exposure for polonium should be based on studies dedicated to squid as well to avoid biased estimations (see below). Also, the ms should have been prepared with more care as there are many mistakes all along 'the text which should have been avoided by a careful reading.

[Figure]

Response: The authors thank the anonymous reviewer for their useful comments. We largely agree with the points raised and revised the manuscript accordingly.

Specific points: The title suggest a general approach on "squid" but only one single species is used in the study. I suggest to modify the title by including the name of the squid as follow "Artificial radionuclides in the neon flying suid Ommastrephes bartramii from. . .."

Response: The title of was changed into "Artificial radionuclides in neon flying squid from northwestern Pacific in 2011 following the Fukushima accident".

Line 10. The correct name of the species is Ommastrephes bartramii. This is to be changed consistently throughout the ms. Also specify here "neon flying squid"

Response: The text was changed throughout to "Ommastrephes bartramii" (same in Line 44). The common name of the species "neon flying squid" was also specified here.

Line 14. It should be easier for the readers to write 2.9 104. This should be used consistently throughout the ms.

Response: In line 14-15, the format of the figure was revised as $2.9 \times 104$. Similar changes were made in Table 2 and Line 123-124

Line 22. I agree that cephalopods constitute an important commercially group but here you considered only one single species and it may be somewhat tricky to extrapolate the present result to the whole group, especially to nectobenthic species (cuttlefishes) and to coastal benthic ones (octopuses). Do you believe that similar results are to be found for such Orders?

Response: We are aware the ERICA-Tool has an option for using transfer parameters from similar species, and of Jeffree et al. 2013 that explores the similarity of transfer parameters among related species. However, this study did not produce sufficient data to test these topics. Therefore, we do not suggest our results be extrapolated to

other species, especially those in highly different environments such as shallow coastal benthic octopus species. Upon review of the text, we found no such extrapolations, including line 22 which simply states that our results add to the scarce data on open-ocean organisms. We compare tissue distribution data against those from another free swimming cephalopod cuttlefish, but take care to avoid a suggestion that squid transfer parameters should be extrapolated to cuttlefish.

Line 32 and Line 38. The years are missing for the references.

Response: Years of the citation were added. The text of references was also updated.

M&M. Where the sexes considered when grouping the individuals? Sexual dimorphism occurs in this species so it can results in grouping individuals of similar size/weight but with different ages. How did you manage this?

Response: (We assume here the question is "Were" (not "Where")). The main purpose of the paper was to report dose rates (to seafood consumers and squid). The study found these doe rates to low relative to benchmarks, and therefore, it was not necessary to explore male/female differences. Although not essential to this study, we agree it is an interesting topic, and could be investigated further in a future study.

Line 72. Gut tissues is very vague and seems to mainly refer to organs and tissues involved in the digestive processes. If this is true, it means that other tissues such as the gills, heart, gonads and associated glands were not considered. Can you please clarify?

Response: "Gut tissue "has now been clarified (lines 85-86).

Line 74. Define HPGe here and remove it at Line 78. Lines 79-80. Detection efficiencies for the other radionuclides should be also provided here. Line 112. "yr-1" . Line 118 and Line 119. Spaces are missing before and inside the references. Please prepare you ms with more care. Line 124. "activity of a radionuclide"

Response: For all of the above, the text was revised accordingly.

Line 127. As for CRWB:water, define CRWB:Tissue

Response: We improved and clarified both descriptions with more information.

Lines 130-140. This paragraph should move to the M&M section: it is not "results" but just a description of the sampling which was missing in the M&M section.

Response: While some of these lines could be moved to the methods section, most of this paragraph is interpretation of data and we prefer the entire paragraph to remain here as it includes discussion and begins a flow of logic that connects to subsequent discussion text.

Page 6. The table is a duplicate of Table 1 page 9. Remove it from page 6.

Response: The table appeared on page 6 by mistake and has been deleted.

Line 147. Do you mean independently of the size classes?

Response: The word "maximum" implies "for all size classes." However, we have added text to clarify.

Line 160. CF factor has been determined experimentally for cuttlefish by Bustamante et al. 2006 in JEMBE with lower values than reported here.

Response: As described above, we have made some tissue distribution comparisons with another cephalopod cuttlefish, but have not compared our open-ocean squid CR data with laboratory-derived cuttlefish CR data. They are two different species, with different diets. But also, laboratory data often under predict CR values due to relatively short exposure times compared with real world conditions, and due to the difficulty of replicating real-world diet pathways in the laboratory. There are multiple factors that can make open-ocean vs laboratory CR data different, as well as the CRs from two species different. While possible, and interesting, such a topic was not in our objectives, and therefore we have chosen to not add a lengthy discussion on an important, but tangential topic. It is a good idea for another paper.

Line 171. Change Bustamante et al 2004 (dedicated to Ag and Co) by Bustamante et al. 2006 (dedicated to Cs and Am).

Response: The range of previous results of Cs in cephalopods was changed into 2–14, with the citation of Bustamante et al 2006.

Line 174. One important aspect is that the digestive gland is the storage tissue independently of the exposure pathway (food or seawater).

Response: We don't disagree. And the point the referee mentions shows a value of laboratory studies where diet vs water exposures can be controlled. But, in this open-ocean study we could not test this question. It would seem somewhat of a reach to include it as a conclusion in this paper.

Line 175. Is this significant?

Response: Yes, it is ($P<0.05$, in t-test).

Line 179. Provide a reference. Line 180. Add Bustamante et al. 2006 as a reference for Cs.

Response: Citation of Bustamante et al., 2004 was changed into Bustamante et al., 2006. Same change in Line 195-196.

Page 9, Table 1. "Statistics" in the title is not appropriate here; there is no statistics in this table but activities of the radionuclides only. For "small individuals", means and standard deviation have been calculated with only 2 individuals, which is not fully correct.

Response: The title of Table 1 was changed into "Radionuclides levels in composite samples". The "n" numbers in this table is the number of composite samples.

Page 11. *** is not applied to Cs, so it should be limited to Ag.

Response: Line 224-225: Text was added to clarify the calculation for the values of

WB-W for Cs-134, Cs-137 and Ag-110m.

Line 216. The value of 15Bq/kg seems a bit high compare to what it is found for muscle in squids. In the cited review (Carvalho 2011) , the value is 1.61 Bq/kg wwt, so I guess you took the wrong value in the table. See also for example Waska et al 2008 in STOTEN who reported 5.7 Bq/kg dry wt (so approx. 5 times less when expressed relatively to the fresh weight) in the squid Todarodes pacificus from the Japan Sea. Also, Heyraud et al. 1994 reported values of 15 to 21 Bq/kg dry wt (so between 3 to 4 in wet wt) in Loligo vulgaris from South Africa. Revise your dose calculation accordingly.

Response: The comment encouraged us to add text that clarifies our approach. The Po-210 value of 15 Bq/kg was selected purposefully. The astute reviewer is correct that it is higher than the average of the available data. As explained in the text, it is being used here as a conservative value in dose calculations. By conservative, we mean it is representative of the upper portion of the available data. This approach is typical in dose assessments. If we used an average value, as suggested, it would ignore the upper 50% of potential dose rates, and could lead to an erroneous result when comparing with benchmarks. We could add dose rates for the average value Po-210, and a low value as well. However, Po-210 is not the focus of the study. It is being presented here simply to provide a context for the FDNPP-related radionuclides, and use of a conservative value is appropriate data for such context. We have clarified the text accordingly.

Line 231. Do you mean "0.010 mSv"?

Response: The figure of "0.01 mSv" was changed into "0.010 mSv" to make the significant digits constant.

Line 234-243. Calculations to be revised according to relevant Po values.

Response: See previous response (two above). For human dose rates, we also do not want to use an average Po-210 value as it is not conservative. Using the average

under predicts 50% of potential dose rates. We use a higher value representative of the upper portion of the data as described above, which is appropriate given we are using the Po-210 dose rate simply for context here. The comment has encouraged us to improve the text on this topic.

References. The bibliographic references should be homogeneous. For example, Line 276, the journal title is not in full as for the other references.

Response: The bibliographic references were updated.

---

## Referee Report (RR1)

Review comments on "Artificial radionuclides in neon flying squid from the northwestern Pacific in 2011 following the Fukushima accident" by Yu et al.

In general, this article will be able to contribute to understand impact on marine biota by Fukushima derived radio activities. The data presented in this article about Ag-110m in neon flying squid is interesting.

1, It is however, a serious question about measurement of Ag-110m activity concentration in seawater which is a base of estimation of CR. The authors report that Ag-110g activity concentrations were below detection limit, but no description how they measure the Ag-110m activity concentration in seawater. In some conditions, Ag-110m can not be extracted well from sea water because Ag-110m exist in organic form. Therefore the authors should state how they measure the Ag-110m activity concentration and need to show reliability of their measurements.

2, Another bog problem in this article is handling manner of the maximum number of significant digit throughout main text and Tables 1 and 2. In general based on the results, the maximum number of significant digit should be two or three, not four or more in case of this article. It is also needed to show the number as consistent number of significant digit throughout the article.

Eg.

$2.95 \times 10^4 \pm 8.94 \times 10^3$ is not correct, this should be $2.95 \times 10^4 \pm 0.89 \times 10^4$ or $(2.95 \pm 0.89) \times 10^4$

$41.87 \pm 39.49$ is not correct, this should be $42 \pm 39$ or $40 \pm 40$

Please check throughout the text and tables.

End of review.

---

## Author Response (AR2)

Referee #4 comments and responses:

In general, this article will be able to contribute to understand impact on marine biota by Fukushima derived radio activities. The data presented in this article about Ag-110m in neon flying squid is interesting.

**Response: The authors thank the anonymous reviewer for their useful comments. We largely agree with the points raised and revised the manuscript accordingly.**

Specific points:

1, It is however, a serious question about measurement of Ag-110m activity concentration in seawater which is a base of estimation of CR. The authors report that Ag-110g activity concentrations were below detection limit, but no description how they measure the Ag-110m activity concentration in seawater. In some conditions, Ag-110m can not be extracted well from sea water because Ag-110m exist in organic form. Therefore the authors should state how they measure the Ag-110m activity concentration and need to show reliability of their measurements.

**Response: One paragraph describing the analytical method for seawater samples was added to the manuscript and the reliability of the method was demostrated.**

2, Another bog problem in this article is handling manner of the maximum number of significant digit throughout main text and Tables 1 and 2. In general based on the results, the maximum number of significant digit should be two or three, not four or more in case of this article. It is also needed to show the number as consistent number of significant digit throughout the article.

**Response: The maximum number of significant digit was changed into 2 in Table 1 and Table 2 and throughout the article main text.**

[revised manuscript text omitted]

---

## Author Response (AR3)

**Comments to the Author:**

Page 3: It is analytically not correct to give detection efficiencies for the geometry used on HPGe detector to 2nd decimal unit for artificial radionuclides (as listed). Is 2.75% different from 2.72%? Efficiencies given as 1.51%, 2.10% and 4.30% are analytically incorrect. Authors should look into associated analytical uncertainties and give detection efficiencies to only 1st decimal unit.

**Response:**

The authors thank the reviewer for the useful comments. We agree with the point raised and revised the manuscript accordingly.